# Assessing the levels of utilisation of WHO-recommended healthcare waste disposal practices at healthcare service delivery points in Uganda

Shafik Senkubuge[1], Lydia Kabwijamu[1], Sarah Nabukeera[2], Andrew K. Tusubira[1], Timothy Kasule[3], Fredrick E. Makumbi[2], Samuel Etajak[1], Christine Nalwadda Kayemba[1]*

1 Department of Community Health & Behavioural Sciences, Makerere University School of Public Health, New Mulago Hospital Complex, Kampala, Uganda, 2 Department of Epidemiology & Biostatistics, Makerere University School of Public Health, New Mulago Hospital Complex, Kampala, Uganda, 3 Reproductive Health Commodity Security, United Nations Population Fund, Nakasero, Kampala, Uganda

* cnalwadda@musph.ac.ug

## Abstract

Inadequate healthcare waste management (HCWM) poses substantial challenges to public health, environmental sustainability and community wellbeing. Safe healthcare waste disposal practices (HWDP) at healthcare facilities are essential to protect staff, patients, caregivers, and the wider public. Achieving this requires consistent use of appropriate final waste disposal practices at each healthcare facility. This study assessed the level of utilisation of World Health Organisation (WHO) - recommended healthcare waste disposal practices at healthcare service delivery points (HSDP) in Uganda. A cross-sectional survey was conducted among 681 randomly selected HSDP from 15 sub-regions of Uganda between September and December 2023. Facilities were categorised into primary, secondary and tertiary levels. The sample was drawn from a national sampling frame of 6,929 HSDP reported through the District Health Information System platform in 2020. At each HSDP, the study assessed the extent to which WHO-recommended HWDP were utilised. Findings showed that over 70% of the surveyed HSDP utilised WHO non-recommended HWDP such as on-site burning, burial, and disposal with general waste. Among primary-level facilities, 30.2% disposed of healthcare waste with regular garbage, 17.3% burnt the waste, and 24.3% buried it within facility grounds. In contrast, 31.7% of secondary-level facilities reported using centralised collection services by specialised agencies. Over half of tertiary facilities (58.3%) disposed of waste with regular garbage, while only 12.5% used incinerators. Regional differences were evident, with disposal alongside general garbage particularly high in Kampala (92.7%) compared to Karamoja (5.3%). Urban HSDP were also more likely to use this method (59.4%) than rural facilities (19.5%). Overall, the nation-wide assessment presents critical gaps in the availability and utilization of WHO-recommended healthcare waste

**Data availability statement:** The dataset generated and analyzed during the study is not publicly available due to confidentiality concerns. However, upon reasonable request, data will be shared by the United Nations Population Fund (UNFPA) Uganda office and stripped of original data identifiers to ensure confidentiality. Only variables used for the analysis will be shared. Data can be requested via email at: hq@unfpa.org.

**Funding:** This study was made possible through funding by the United Nations Population Fund (UNFPA) Supplies Partnership Programme to CNK. UNFPA experts guided the research team in the selection of the study design, rolling out of the sampling framework, use of systmapp application for data collection, and validation of the manuscript. However, the views expressed do not necessarily reflect the UNFPA's official policies.

**Competing interests:** The authors have declared that no competing interests exist.

disposal practices across HSDP levels, management types, residence settings, and regions. Addressing these gaps requires targeted interventions, including strengthening guideline enforcement, improving infrastructure, and enhancing compliance with healthcare waste management regulations.

## Background

Global concerns over healthcare waste management have increased in recent years due to its potential risks to the environment and public health. It is widely recognised that healthcare activities inherently generate waste, which, if not managed properly, poses serious threats to human health, the environment, and community safety [1,2]. Ineffective waste management practices can lead to healthcare-associated infections (HAIs), occupational hazards, and contamination of food and water sources. For example, mishandled sharps and other contaminated waste facilitate the spread of infectious diseases like hepatitis B, hepatitis C, and HIV/AIDS, endangering healthcare workers, patients, waste handlers, and the general public [3,4]. HCF support staff, visitors and the general public are more vulnerable to these infections [5,6].

The World Health Organization defines healthcare waste as all waste produced within healthcare facilities, such as hospitals, health centres, and pharmaceutical shops [6]. The literature highlights that approximately 80% of waste generated in healthcare settings is non-infectious and suitable for inclusion in the municipal waste stream, while the remaining 20% is classified as infectious waste requiring specialised treatment and disposal methods [7–9]. The use of WHO-recommended HCWM practices, such as segregation of HCW using colour-coded bins with liners, use of eco-friendly incinerators, and disposal by approved central agencies, are some of the practices that have been documented by [10,11], which can mitigate health and environmental risks posed by HCW [12].

WHO further discourages open burning and instead promotes environmentally friendly treatment alternatives. These include; autoclaving (steam treatment), which uses pressurised steam at high temperatures to disinfect waste; microwaving, which applies radiant energy to destroy pathogens; and chemical disinfection, particularly for liquid waste. Incineration remains an option where appropriate, but it must be conducted in well-designed and properly operated incinerators that burn at temperatures above 800°C to ensure complete combustion and to minimise the release of harmful pollutants such as dioxins and furans [12].

Environmental pollution linked to inadequate healthcare waste management contributed to 23% of global deaths, corresponding to 12.6 million deaths and 22% of disability-adjusted life years (DALYs) in 2012 [13]. Despite these appalling figures, healthcare waste management has not received adequate attention, especially in low- and middle-income countries, particularly in Africa [14,15]. Limited funding, inefficient waste segregation systems, and gaps in knowledge and practice in final treatment and disposal make it challenging for facilities in these regions to consistently follow WHO-recommended waste management practices [16,17]. Consequently,

healthcare facilities employ a mix of waste disposal methods, with varying adherence to WHO-recommended practices [18–20].

In Uganda, HSDP generate substantial volumes of waste, with hospitals averaging 92 kilograms daily, while primary healthcare facilities such as Health Centre (HC) IV, HC III, and HC II generate approximately 42 kg, 25 kg, and 20 kg daily, respectively [21]. Despite policies and guidelines set by the Ugandan Ministry of Health for injection safety and healthcare waste management, many HSDPs still struggle with proper healthcare waste-handling and disposal practices [22–24]. While various studies have examined waste management practices in Uganda, most have focused on high-level healthcare facilities in specific regions, leaving a gap in comprehensive data across all facility levels [18,24,25].

This paper presents findings on the levels of utilisation of WHO-recommended healthcare waste disposal practices in Uganda, providing critical insights that can inform policy improvements and targeted interventions for safe and sustainable healthcare service delivery.

## Methods

### Study setting

The study was conducted between September and December 2023 in 15 UBOS statistical sub-regions of Uganda, which include; Kampala, South Buganda, North Buganda, Busoga, Bukedi, Bugisu, Teso, Karamoja, Lango, Acholi, West Nile, Bunyoro, Tooro, Kigezi, and Ankole [26].

### Study design

A descriptive cross-sectional study design in which quantitative data collection techniques were utilised.

### Study population

This study was part of the larger assessment of health service delivery points for reproductive health commodities in Uganda. The study population included health facilities referred to as HSDP. We included both private and public HSDP at different levels, such as Clinics, Health Centre III, Health Centre IV, and General Hospitals (GH), Regional Referral Hospitals (RRH) and National Referral Hospitals (NRH), which constituted the HSDP for modern methods of contraceptives and reproductive health services in 112 districts of Uganda.

In line with the standard UNFPA methodology for the HSDP surveys, the healthcare facilities were categorised into three broad categories, namely; primary, secondary and tertiary HSDP. The primary HSDP comprised of clinics, HCIIs and HCIIIs; secondary HSDP comprised of HCIV and General Hospitals, and Tertiary-level care HSDP comprised of Regional Referral Hospitals and National Referral Hospitals.

### Sample size and sampling procedures

The sample size of the HSDP was determined using the standard UNFPA sampling methodology for the HSDP survey, which was adapted to the Ugandan context. HSDPs at the different levels were selected using the Ministry of Health (MOH) 2022 master list, which provided 7144 HSDPs. The sample size was estimated using the Leslie Kish formula [27], at a confidence interval of 1.96, at 5% level and a permissible error of 0.05 and a relative proportion, p, for the level of the facility, non-response rate of 10.6%, resulting in 681 HSDP.

Probability proportionate to size was then used to determine the number of HSDPs to include per region and level. Lastly, simple random sampling was used to select districts and HSDP to be assessed in each of the 15 regions by category, that is, Tertiary, Secondary and Primary levels.

## Data collection

Data collection was conducted from 14/09/2023–19/11/2023. A face-to-face interviewer-administered questionnaire provided by UNFPA was used to collect quantitative data, to capture the health facilities' characteristics and healthcare waste disposal practices. The areas of healthcare waste disposal assessed included: incinerators, healthcare waste collection bins, burial grounds and central agencies used in the disposal of healthcare waste at the HSDP.

Data was collected electronically using the Systmapp application. The systmapp application is a web-based data collection application developed by UNFPA that is accessed at https://supplies.systmapp.com/Account/Login. Research assistants were required to download the app, sign in using a system-generated password and collect data using mobile phones. Research assistants collected data offline and would upload it daily once they accessed the internet.

Before data collection, all research assistants were trained on the study objectives, methodology and how to use the systmapp application. The research assistants also participated in the pilot of the data collection tools. During data collection, the system administrator and data manager would review data for completeness daily and give feedback to the field teams.

## Study outcomes

This paper focuses on the levels of utilisation of the WHO-recommended HWDP at the HSDP. This was measured using responses to specific questions in the data collection tool that was administered to the HSDP managers (HSDP incharges, clinicians and IPC focal point persons). These responded with "Yes" or "No" to the use of an ecofriendly incinerators, use of a specific healthcare waste disposal agency, use of healthcare waste collection bins and burial of healthcare waste on grounds of the HSDP. Those HSDP that reported the use of the WHO-recommended HWDP were recorded and compared with those that used the WHO non-recommended HWDP across the HSDP levels, residence status, management type and regions.

## Data management and analysis

The Data manager and IT administrator downloaded the data from the server and exported it to the STATA program for analysis. Data analysis was done using version STATA 15. Descriptive analyses were done to generate frequencies, percentages, means and summations.

## Ethics approval and consent to participate

Ethical approval was obtained from the Makerere University School of Public Health Research Ethics Committee (SPH-2023–455). This study was registered with the Uganda National Council of Science and Technology (HS 3131ES). Before conducting the study, administrative clearances were obtained from MOH and the respective District Local governments. The research team obtained written informed consent from participants before all data collection activities.

## Results

A total of 681 of 762 HSDP were surveyed, resulting in a response rate of 89.4%. Disaggregated by the level of service delivery, the majority of the HSDP 84.14% (573/681) were at the primary level. With the largest proportion of the HSDP, 35.24% (240/681), located in Kampala, the Central region. A disproportionate distribution of HSDP was observed across the Northern, Western, and Eastern regions, with Ankole and Busoga sub-regions having the highest number of HSDP surveyed, 8.81% (60/681) and 7.64% (52/681), respectively. The majority of HSDP, 62.41% (425/681), were located in rural areas and over half of the HSDP, 55.51% (378/681), were managed by the government, as presented in Table 1.

## HWDP exhibited by various HSDP across regions

The majority of the HSDP in Kampala, 92.7% (76/82), disposed of healthcare waste with regular garbage, unlike 5.3% (1/19) of HSDP in Karamoja, which exhibited the same practice. Over half of HSDP in West Nile, 56.5% (26/46), reported that waste was centrally collected by specific agencies for disposal away from the HSDP.

**Table 1. Shows the characteristics of HSDP by facility level, location, residences and management status.**

| Characteristics | | Frequency (n) |
|---|---|---|
| **Type of Facility** | Primary level | 573 |
| | Secondary level | 60 |
| | Tertiary level | 48 |
| **Region** | | |
| **North** | Acholi | 31 |
| | Karamoja | 19 |
| | Lango | 23 |
| | West Nile | 46 |
| **West** | Ankole | 60 |
| | Bunyoro | 35 |
| | Kigezi | 37 |
| | Tooro | 51 |
| **East** | Bugisu | 34 |
| | Bukedi | 27 |
| | Busoga | 52 |
| | Teso | 26 |
| **Central** | Kampala | 82 |
| | North Central | 82 |
| | South Central | 76 |
| **Residence** | Rural | 425 |
| | Urban | 251 |
| **Management** | Government | 378 |
| | Private | 207 |
| | NGO/PNFP | 49 |
| | Others | 42 |
| **Total** | | **681** |

In Bukedi region, the majority of the 27 HSDP 44.4% (12/27) reported that they used incinerators for the final disposal of waste. However, Acholi, Karamoja, Lango, Ankole, Kigezi, and Tooro regions did not have incinerators. In Teso region, half of the HSDP 50.0% (13/26) reported that they bury waste in special dump pits on the grounds of the HSDP and 43.5% (23/43) of the HSDP in Lango reported burning waste on the grounds of the HSDP, as presented in Table 2.

**Utilisation of WHO-recommended healthcare waste disposal practices at HSDP in Uganda**

The study findings portray varying levels of use of WHO-recommended healthcare waste disposal practices across HSDP levels, by geographical locations, management status and residence status. Overall, only 30% (204/681) of the surveyed HSDP reported the use of WHO-recommended healthcare waste disposal practices; use of incinerators and central collection by a specific agency for disposal away from the HSDP as presented in Fig 1.

The use of WHO-recommended healthcare waste disposal practices was more prevalent at secondary and tertiary HSDP as compared to primary level HSDP. The majority 31.7% (19/60) of the secondary-level HSDP used centralised collection by specialised agencies and 12.5% (6/48) of the tertiary-level HSDP reported the use of incinerators. In contrast, primary-level HSDP primarily used WHO non-recommended practices such as on-site burning (17.3%, 99/573) and burying waste in dedicated dump pits (24.3%, 139/573) as highlighted in Fig 2.

**Table 2. Portraying the level, management status, residence and geographical location of the HSDP and the variation of HWDP.**

| Characteristics | Frequency | Healthcare waste disposal practices reported in percentage (%) | | | | |
|---|---|---|---|---|---|---|
| | | Burning on the grounds of the HSDP | Bury in special dump pits on the grounds of the HSDP | Disposed of with regular garbage | Use of Incinerators | Centrally collected by a specific agency for disposal away from the HSDP |
| **Overall** | **681** | **14.8** | **20.7** | **34.5** | **9.1** | **20.9** |
| **Type of Facility** | | | | | | |
| Primary level | 573 | 17.3 | 24.3 | 30.2 | 9.3 | 19 |
| Secondary level | 60 | 3.3 | 3.3 | 56.7 | 5 | 31.7 |
| Tertiary level | 48 | 0 | 0 | 58.3 | 12.5 | 29.2 |
| **Region** | | | | | | |
| **North** | | | | | | |
| Acholi | 31 | 29 | 3.2 | 25.8 | 0 | 41.9 |
| Karamoja | 19 | 21.1 | 31.6 | 5.3 | 0 | 42.1 |
| Lango | 23 | 43.5 | 34.8 | 21.7 | 0 | 0 |
| West Nile | 46 | 17.4 | 10.9 | 10.9 | 4.4 | 56.5 |
| **West** | | | | | | |
| Ankole | 60 | 30 | 10 | 40 | 0 | 20 |
| Bunyoro | 35 | 5.7 | 22.9 | 34.3 | 2.9 | 34.3 |
| Kigezi | 37 | 18.9 | 37.8 | 29.7 | 0 | 13.5 |
| Tooro | 51 | 13.7 | 25.5 | 31.4 | 0 | 29.4 |
| **East** | | | | | | |
| Bugisu | 34 | 8.8 | 47.1 | 14.7 | 20.6 | 8.8 |
| Bukedi | 27 | 18.5 | 14.8 | 14.8 | 44.4 | 7.4 |
| Busoga | 52 | 15.4 | 48.1 | 15.4 | 1.9 | 19.2 |
| Teso | 26 | 0 | 50 | 3.9 | 19.2 | 26.9 |
| **Central** | | | | | | |
| Kampala | 82 | 0 | 0 | 92.7 | 3.7 | 3.7 |
| North Central | 82 | 8.5 | 14.6 | 37.8 | 25.6 | 13.4 |
| South Central | 76 | 17.1 | 13.2 | 36.8 | 13.2 | 19.7 |
| **Residence** | | | | | | |
| Rural | 425 | 19.1 | 28.7 | 19.5 | 11.1 | 21.7 |
| Urban | 251 | 8 | 7.2 | 59.4 | 5.6 | 19.9 |
| **Management** | | | | | | |
| Government | 378 | 21.2 | 28.6 | 22.8 | 7.4 | 20.1 |
| Private | 207 | 4.8 | 9.7 | 58 | 10.6 | 16.9 |
| NGO/PNFP | 49 | 8.2 | 18.4 | 24.5 | 16.3 | 32.7 |
| Others | 42 | 16.7 | 7.1 | 33.3 | 7.1 | 35.7 |

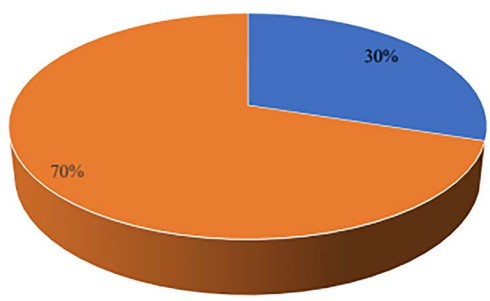

**Fig 1. Level of utilization of WHO-recommended healthcare waste disposal practices at HSDPs in Uganda.**

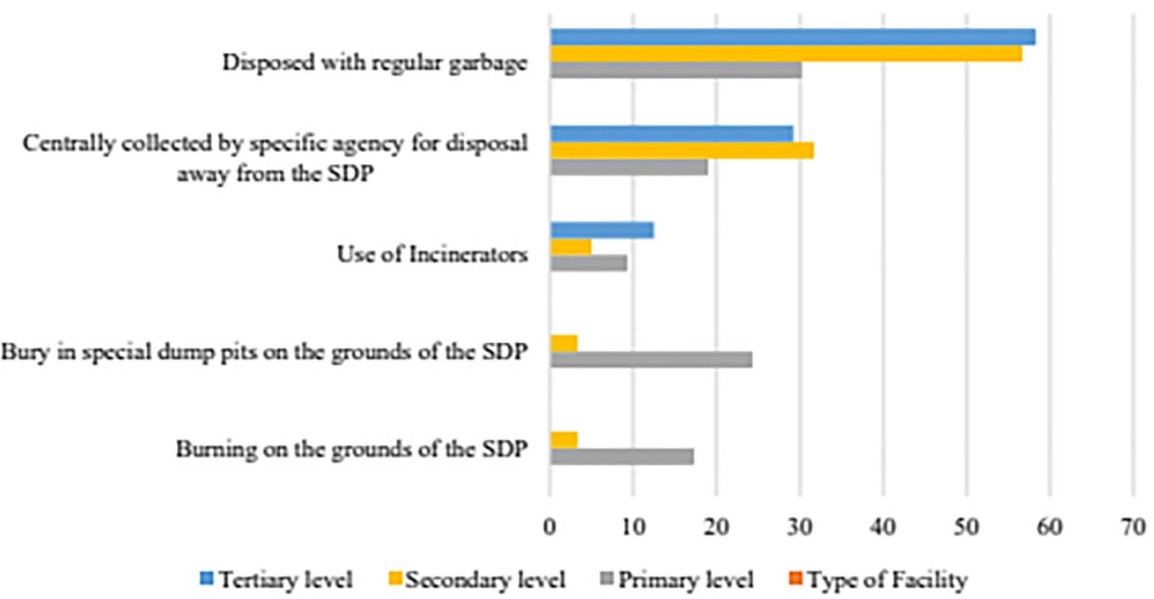

**Fig 2. Variation of health care waste disposal practice across the HSDP levels.**

Alarmingly, over half, 58.3% (28/48) of tertiary and 56.7% (34/60) of the secondary-level HSDP used -WHO non-recommended disposal practices; disposing of healthcare waste with regular garbage (Table 3).

## HWDP at the HSDP in the urban and rural residences

The majority of the HSDP in urban residences 59.4% (149/251), disposed of waste in regular garbage bins and 28.7% (122/425) carried out the burying of waste in special dump pits on the grounds of the HSDP. However, for the HSDP in the rural residence, only 19.1% (81/425), reported burning waste on the grounds of the HSDP.. With the majority of the HSDP 21.7% (92/425), reporting that healthcare waste is centrally collected by specific agencies for disposal away from the HSDP.

**Table 3. Shows the levels of utilization of WHO-recommended healthcare waste disposal practices across the various HSDP levels, regions, residence and management status.**

| Characteristics | Frequency (n) | Healthcare waste disposal practices at HSDP | | |
| --- | --- | --- | --- | --- |
| | | WHO non-recommended HWDP; Burning, burying and disposal with regular garbage | WHO recommended HWDP; Use of incinerators and centrally collected by a specific agency for disposal | Total |
| **Overall** | **681** | **70** | **30** | **100** |
| **Type of HSDP** | | | | |
| Primary level | 573 | 71.8 | 28.3 | 100 |
| Secondary level | 60 | 63.3 | 36.7 | 100 |
| Tertiary level | 48 | 58.3 | 41.7 | 100 |
| **Region** | | | | |
| **North** | | | | |
| Acholi | 31 | 58 | 41.9 | 100 |
| Karamoja | 19 | 58 | 42.1 | 100 |
| Lango | 23 | 100 | 0 | 100 |
| West Nile | 46 | 39.2 | 60.9 | 100 |
| **West** | | | | |
| Ankole | 60 | 80 | 20 | 100 |
| Bunyoro | 35 | 62.9 | 37.2 | 100 |
| Kigezi | 37 | 86.4 | 13.5 | 100 |
| Tooro | 51 | 70.6 | 29.4 | 100 |
| **East** | | | | |
| Bugisu | 34 | 70.6 | 29.4 | 100 |
| Bukedi | 27 | 48.1 | 51.8 | 100 |
| Busoga | 52 | 78.9 | 21.1 | 100 |
| Teso | 26 | 53.9 | 46.1 | 100 |
| **Central** | | | | |
| Kampala | 82 | 92.7 | 7.4 | 100 |
| North Central | 82 | 60.9 | 39 | 100 |
| South Central | 76 | 67.1 | 32.9 | 100 |
| **Residence** | | | | |
| Rural | 425 | 67.3 | 32.8 | 100 |
| Urban | 251 | 74.6 | 25.5 | 100 |
| **Management** | | | | |
| **Government** | 378 | 72.6 | 27.5 | 100 |
| Private | 207 | 72.5 | 27.5 | 100 |
| NGO/PNFP | 49 | 51.1 | 49 | 100 |
| Others | 42 | 57.1 | 42.8 | 100 |

## Discussion

This study presents findings from a nationwide assessment of the levels of utilisation of the WHO-recommended HWDP at HSDP in Uganda. More than 70% of the surveyed HSDP used WHO non-recommended HCW disposal methods. These include; on-site burning, burying, and disposal of healthcare waste with general waste. This portrays systemic weaknesses and regional disparities in appropriate HCWD. For instance, most of the HSDP in Kampala, the capital city, disposed of HCW with regular waste and the majority of the HSDP in densely populated regions such as Kigezi, Tooro

and Acholi did not have incinerators for final HCW disposal. These findings provide a glance at the status of use of WHO-recommended HWDP practices at HSDP in Uganda [28].

The predominant use of WHO non-recommended HWDP observed at HSDP in Uganda mimics findings from other Low-and Middle income countries [24]. Studies in Tanzania and Kenya reported that healthcare facilities, especially at the primary level, lacked appropriate infrastructure, technical capacity, and funding for proper healthcare waste disposal [20,25,29]. Previous assessments conducted in HCFs in Uganda attributed poor HCWD practices to limited resources, inadequate technical skills, and weak planning frameworks, which likely contributed to the results observed in this study [24].

The majority of the lower-level HSDP practised open burning and burial of healthcare waste on the grounds of the facility. These findings are similar to those from Tanzania and Ethiopia, where such practices were linked to risks of HAIs, needle-stick injuries, and environmental contamination [20,25,30]. Open burning of HCW without temperature regulation or pollution control has been shown to aerosolise viable pathogens, exposing nearby residents, patients, and healthcare workers to infectious and toxic substances. In addition, incomplete combustion releases dioxins, furans, and particulate matter, which persist in the environment and bioaccumulate through the food chain, posing long-term health and ecological hazards [31]. Likewise, mixing healthcare waste with municipal waste increases risks of exposure for waste collectors and nearby communities while contaminating groundwater sources and attracting disease vectors such as flies and rodents [30].

Although most tertiary HSDP reported having incinerators, a majority still engaged in non-WHO-recommended HWDP. This finding aligns with studies from Kenya and Nigeria showing that while higher-level facilities may have better infrastructure, the limited functionality and maintenance of incinerators often lead to the adoption of unsafe alternatives [29,32]. The uneven distribution of waste management infrastructure, especially between urban and rural areas, continues to impede the full implementation of the WHO-recommended HWDP. These challenges are echoed in other low-resource countries [14,15,17,20]

Substantial regional and institutional disparities were observed. For example, Bukedi region reported relatively high utilisation of incinerators, while most of the HSDPs in Acholi and Karamoja did not have any such infrastructure by the time of the survey. Studies from Ethiopia and Ghana have similarly shown that unequal allocation of waste management resources results in inconsistent and often substandard HWDP outcomes [17,20]. In Kampala, most HSDP disposed of HCW in regular garbage bins despite expectations of better infrastructure - a contrast to findings from urban centres in other African countries where the availability of equipment and resources supported better practices [19]. These disparities highlight the need for context-specific interventions that address regional infrastructure and resource gaps.

In terms of management type, privately owned and NGO-run HSDP generally exhibited better compliance with WHO-recommended HWDP than public facilities. Comparable results have been documented in other countries [33,34], where private facilities benefit from external funding, better supervision, and stricter adherence to protocols [17]. This highlights the need for the Government of Uganda to enhance support and supervision for public facilities to ensure uniform compliance and quality across all service providers.

## Strengths

This study's primary strength is its nationwide coverage, covering all 15 UBOS statistical regions of Uganda. By including a representative sample of 681 HSDPs across primary, secondary, and tertiary levels, the study captures a detailed and accurate picture of the levels of utilisation of WHO-recommended healthcare waste disposal practices across different HSDP levels and geographic contexts. Furthermore, the application of the UNFPA's standardised sampling methodology enhances the reliability of findings, providing robust data that can inform national policy and targeted interventions in healthcare waste management.

## Limitations

Despite the nationwide coverage, the study has several limitations. First, data collection relied on self-reported data collection tools at HSDP, which might have introduced response bias, potentially affecting the accuracy of the reported levels of utilization of WHO-recommended HCWM practices. The data missed certain focus areas of healthcare waste management which could not be reported on. But, the available data provides insight into the utilization of the WHO -recommended healthcare waste disposal practices at HSDP in Uganda. The IPC audits require triangulation with an observational checklist. The study suggest that a future mixed-methods study with qualitative and observational data collection should be conducted for validation.

Despite the study assessing the levels of utilisation of WHO-recommended HWDP, we did not differentiate between the various types of healthcare waste, for instance, infectious and non-infectious, yet it is vital in determining the appropriateness of the HWDP used by the healthcare facilities.

## Conclusion

This study provides crucial insights into the nationwide utilisation of WHO-recommended healthcare waste disposal practices across Uganda's HSDP, highlighting regional and facility-level disparities. While tertiary HSDP and certain regions show higher utilisation of practices such as incineration and centralised disposal. Primary-level facilities and many rural regions face substantial barriers to implementing these methods, often resorting to WHO non-recommended practices. These findings underscore the need for region-specific infrastructure improvements and enhanced government support to standardise healthcare waste management practices across all HSDPs. Addressing these challenges will be essential to minimising healthcare risks linked to poor HWDP, reducing environmental pollution, and ensuring the safe and effective disposal of healthcare waste throughout Uganda.

## Author contributions

**Conceptualization:** Shafik Senkubuge, Lydia Kabwijamu, Sarah Nabukeera, Andrew K Tusubira, Samuel Etajak, Christine Nalwadda Kayemba.

**Data curation:** Sarah Nabukeera.

**Formal analysis:** Lydia Kabwijamu, Sarah Nabukeera, Fredrick E. Makumbi, Christine Nalwadda Kayemba.

**Funding acquisition:** Timothy Kasule.

**Investigation:** Lydia Kabwijamu, Sarah Nabukeera, Fredrick E. Makumbi.

**Methodology:** Lydia Kabwijamu, Sarah Nabukeera, Fredrick E. Makumbi.

**Project administration:** Lydia Kabwijamu, Timothy Kasule.

**Resources:** Timothy Kasule, Christine Nalwadda Kayemba.

**Supervision:** Christine Nalwadda Kayemba.

**Validation:** Timothy Kasule, Samuel Etajak, Christine Nalwadda Kayemba.

**Visualization:** Samuel Etajak, Christine Nalwadda Kayemba.

**Writing – original draft:** Shafik Senkubuge, Andrew K Tusubira, Samuel Etajak, Christine Nalwadda Kayemba.

**Writing – review & editing:** Shafik Senkubuge, Lydia Kabwijamu, Sarah Nabukeera, Andrew K Tusubira, Fredrick E. Makumbi, Samuel Etajak, Christine Nalwadda Kayemba.

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
