## [Decision Letter · Decision Letter 0]

15 May 2025

PGPH-D-25-00169

Magnitude of utilization of recommended healthcare waste management practices at healthcare service delivery points in Uganda

Dear Dr. Kayemba,

Thank you for submitting your manuscript to PLOS Global Public Health. After careful consideration, we feel that it has merit but does not fully meet PLOS Global Public Health’s publication criteria as it currently stands. Therefore, we invite you to submit a revised version of the manuscript that addresses the points raised during the review process.

Your manuscript has been edited by two reviewers with different expertise, both of whom have provided extensive feedback on your manuscript. I would ask you to carefully review the comments and revise your manuscript accordingly, providing a point-by-point response to the reviewers upon resubmission.

We look forward to receiving your revised manuscript.

Kind regards,

Sarah Jose, Ph.D.

Staff Editor

Journal Requirements:

Additional Editor Comments (if provided):

Reviewers' comments:

Reviewer's Responses to Questions

**Comments to the Author**

1. Does this manuscript meet PLOS Global Public Health’s publication criteria? Is the manuscript technically sound, and do the data support the conclusions? The manuscript must describe methodologically and ethically rigorous research with conclusions that are appropriately drawn based on the data presented.? Is the manuscript technically sound, and do the data support the conclusions? The manuscript must describe methodologically and ethically rigorous research with conclusions that are appropriately drawn based on the data presented.

Reviewer #1: Yes

Reviewer #2: Partly

2. Has the statistical analysis been performed appropriately and rigorously?

Reviewer #1: Yes

Reviewer #2: No

3. Have the authors made all data underlying the findings in their manuscript fully available (please refer to the Data Availability Statement at the start of the manuscript PDF file)?

The PLOS Data policy requires authors to make all data underlying the findings described in their manuscript fully available without restriction, with rare exception. The data should be provided as part of the manuscript or its supporting information, or deposited to a public repository. For example, in addition to summary statistics, the data points behind means, medians and variance measures should be available. If there are restrictions on publicly sharing data—e.g. participant privacy or use of data from a third party—those must be specified.requires authors to make all data underlying the findings described in their manuscript fully available without restriction, with rare exception. The data should be provided as part of the manuscript or its supporting information, or deposited to a public repository. For example, in addition to summary statistics, the data points behind means, medians and variance measures should be available. If there are restrictions on publicly sharing data—e.g. participant privacy or use of data from a third party—those must be specified.

Reviewer #1: Yes

Reviewer #2: Yes

4. Is the manuscript presented in an intelligible fashion and written in standard English?

Reviewer #1: Yes

Reviewer #2: No

5. Review Comments to the Author

Reviewer #1: This manuscript addresses an important and under-explored topic within healthcare systems in low- and middle-income countries: the real-world implementation of healthcare waste management (HCWM) practices across a wide spectrum of health service delivery points (HSDPs). The topic is highly relevant to public health, environmental safety, and infection prevention and control (IPC). The national scope and robust sample design are major strengths. However, certain methodological and analytical aspects need to be refined to strengthen the paper’s utility for both academic and policy audiences.

This manuscript presents important findings that deserve publication but requires clarification of terminology, stronger alignment with IPC frameworks, and enhanced discussion of health risks and policy implications to fully realize its value.

While the dataset is rich and nationally representative, the interpretation and discussion sections would benefit from deeper engagement with IPC frameworks, risk stratification of waste types, and implications for nosocomial infections and occupational health.

The study discusses "recommended" vs. "non-recommended" HCWM practices without distinguishing waste types (e.g., sharps, infectious, pharmaceutical). From an IPC standpoint, the risk profile and required disposal methods vary significantly. I recommend to Include classification of waste types and tailor the evaluation of practices accordingly (e.g., whether sharps were appropriately segregated and disposed of).

The binary categorization of practices as recommended or non-recommended oversimplifies the complexities of IPC-compliant waste handling. Incorporate a graded compliance assessment based on core IPC principles such as segregation at point-of-care, use of color-coded bins, interim storage, and final disposal.

The manuscript does not mention whether the HSDPs assessed had IPC committees or focal persons, which are crucial in overseeing HCWM practices Add a variable (if available) or discuss the impact of functional IPC programs on adherence to recommended HCWM.

The discussion should elaborate more explicitly on how specific non-recommended practices (e.g., open burning, disposal with general garbage) contribute to increased risk of healthcare-associated infections (HAIs), needle-stick injuries, and environmental contamination.

The reliance on self-reported data is a major limitation. IPC audits typically require triangulation with observational checklists. Acknowledge the potential bias more strongly and suggest a future mixed-methods study with observational data collection for validation.

Explicitly state which guidelines were used as the benchmark and how local adaptations may differ.

o The discussion could be improved by more explicitly linking findings to known risks of healthcare-associated infections (HAIs), occupational exposure, and environmental contamination—particularly in relation to practices such as open burning or co-disposal with general waste.

Clarify if “incinerators” include modern controlled combustion systems or rudimentary pits, as this affects emission risks.

Correct typographical inconsistencies (e.g., “healthcare waste” is sometimes inconsistently used with “medical waste”).

Consider adding maps or visual dashboards showing regional disparities in compliance rates for better visual impact.

Reviewer #2: The manuscript title indicates its focus on medical waste management practices. This however is misleading as the research is limited to the examination of options for final disposal. A more appropriate topic would read; A nationwide assessment of healthcare waste disposal in Uganda.

Given that the topic is inappropriate, the literature review does not match the reported results and findings.

Lines 82-85: Incorrect assertion.

The approach used in collecting data from the questionnaire is unclear. In the area of competing interests, reported self-reporting data collection and face-to-face interviews was identified, in Line 143

In terms of results presentation, the information contained in the tables (Table 1) doesn't tally with that in the text. The values in Table 1 does not add up.... 681, 676.There is also some confusion between 681 and 682. This shortcoming is a result of a poor design concepts of the Tables. I suggest two separate tables be conceived.. one dedicated to the characteristics of the facilities in terms of geographical distribution, setting (urban/rural) and ownership status, a second table for reporting the choice and frequencies of final disposal methods.

Finally, I think the study lacks depth. The authors may consider introducing other aspects that will give some insights that will improve the quality and usefulness of the research.

6. PLOS authors have the option to publish the peer review history of their article (what does this mean?). If published, this will include your full peer review and any attached files.). If published, this will include your full peer review and any attached files.

**Do you want your identity to be public for this peer review?** For information about this choice, including consent withdrawal, please see our Privacy Policy..

Reviewer #1: No

Reviewer #2: **Yes:**MANGA, Veronica EbotMANGA, Veronica Ebot

---

## [Decision Letter · Decision Letter 1]

4 Sep 2025

PGPH-D-25-00169R1

Magnitude of utilisation of recommended healthcare waste disposal practices at healthcare service delivery points in Uganda

Dear Dr. Kayemba,

Thank you for submitting your manuscript to PLOS Global Public Health. After careful consideration, we feel that it has merit but does not fully meet PLOS Global Public Health’s publication criteria as it currently stands. Therefore, we invite you to submit a revised version of the manuscript that addresses the points raised during the review process.

Your manuscript has been evaluated by Reviewer 2 of the previous round of peer review, and their comments are appended below.

The reviewer has identified remaining concerns, particularly relating to revision of Table 2 for clarity, but also regarding statements requiring support by appropriate literature references, and the structure of the discussion section.

Additionally, the comments from Reviewer 1 from the previous round of peer review do not appear to have been clearly and comprehensively addressed. As such, please submit a Response to Reviewers file with your revised manuscript that addresses both 1) the comments from Reviewer 2 below and 2) the comments from Reviewer 1 from the previous round of review, and ensure that the responses are arranged by comment and not by manuscript section. Please note that if these comments are not all clearly addressed in further revisions then you submission may be rejected.

We look forward to receiving your revised manuscript.

Kind regards,

Hugh Cowley

Staff Editor

Journal Requirements:

Additional Editor Comments (if provided):

Reviewers' comments:

Reviewer's Responses to Questions

**Comments to the Author**

1. If the authors have adequately addressed your comments raised in a previous round of review and you feel that this manuscript is now acceptable for publication, you may indicate that here to bypass the “Comments to the Author” section, enter your conflict of interest statement in the “Confidential to Editor” section, and submit your "Accept" recommendation.

Reviewer #2: (No Response)

2. Does this manuscript meet PLOS Global Public Health’s publication criteria? Is the manuscript technically sound, and do the data support the conclusions? The manuscript must describe methodologically and ethically rigorous research with conclusions that are appropriately drawn based on the data presented.? Is the manuscript technically sound, and do the data support the conclusions? The manuscript must describe methodologically and ethically rigorous research with conclusions that are appropriately drawn based on the data presented.

Reviewer #2: Yes

3. Has the statistical analysis been performed appropriately and rigorously?

Reviewer #2: Yes

4. Have the authors made all data underlying the findings in their manuscript fully available (please refer to the Data Availability Statement at the start of the manuscript PDF file)?

The PLOS Data policy requires authors to make all data underlying the findings described in their manuscript fully available without restriction, with rare exception. The data should be provided as part of the manuscript or its supporting information, or deposited to a public repository. For example, in addition to summary statistics, the data points behind means, medians and variance measures should be available. If there are restrictions on publicly sharing data—e.g. participant privacy or use of data from a third party—those must be specified.requires authors to make all data underlying the findings described in their manuscript fully available without restriction, with rare exception. The data should be provided as part of the manuscript or its supporting information, or deposited to a public repository. For example, in addition to summary statistics, the data points behind means, medians and variance measures should be available. If there are restrictions on publicly sharing data—e.g. participant privacy or use of data from a third party—those must be specified.

Reviewer #2: Yes

5. Is the manuscript presented in an intelligible fashion and written in standard English?

Reviewer #2: No

6. Review Comments to the Author

Reviewer #2: 1. In my previous review, I suggested the revision of Table 2 for more clarity. The present submission has not met that requirement. The last row of Table 2 indicates Total... whereas the figures above in the respective columns do not add up to those figures i.e. 681 14.8 20.7 34.5 9.1 20.9. One of the ways to address this would be to separate the tables. Results for the 15UBOs could be presented on a separate Table where frequencies (n) and % can be included and for which total values would be the that appears on the last row is a summation of values in that particular column.

2. Citation of '1965' in line 44 is not in Reference section

3. Delete 'below' in line 237

4. Lack of clarity of lines 267 to 268

5. Claims not supported by previous findings lines 267 to 275; 283 - 289.

6. Discussion section needs restructuring: There is poor paragraphing, inappropriate use of proper linking sentences and logical connections that limits readability.

7. PLOS authors have the option to publish the peer review history of their article (what does this mean?). If published, this will include your full peer review and any attached files.). If published, this will include your full peer review and any attached files.

**Do you want your identity to be public for this peer review?** For information about this choice, including consent withdrawal, please see our Privacy Policy..

Reviewer #2: No

---

## [Decision Letter · Decision Letter 2]

1 Feb 2026

PGPH-D-25-00169R2

Magnitude of utilisation of recommended healthcare waste disposal practices at healthcare service delivery points in Uganda

Dear Dr. Kayemba,

Thank you for submitting your manuscript to PLOS Global Public Health. After careful consideration, we feel that it has merit but does not fully meet PLOS Global Public Health’s publication criteria as it currently stands. Therefore, we invite you to submit a revised version of the manuscript that addresses the points raised during the review process.

The manuscript has been evaluated by a third reviewer, and their comments are available below.

The reviewer has raised a number of concerns that need attention. They request some clarification of terminology, particularly around statistical significance.

Could you please revise the manuscript to carefully address the concerns raised?

We look forward to receiving your revised manuscript.

Kind regards,

Jen Edwards

Staff Editor

Journal Requirements:

Additional Editor Comments (if provided):

Reviewers' comments:

Reviewer's Responses to Questions

**Comments to the Author**

1. If the authors have adequately addressed your comments raised in a previous round of review and you feel that this manuscript is now acceptable for publication, you may indicate that here to bypass the “Comments to the Author” section, enter your conflict of interest statement in the “Confidential to Editor” section, and submit your "Accept" recommendation.

Reviewer #3: (No Response)

2. Does this manuscript meet PLOS Global Public Health’s publication criteria? Is the manuscript technically sound, and do the data support the conclusions? The manuscript must describe methodologically and ethically rigorous research with conclusions that are appropriately drawn based on the data presented.? Is the manuscript technically sound, and do the data support the conclusions? The manuscript must describe methodologically and ethically rigorous research with conclusions that are appropriately drawn based on the data presented.

Reviewer #3: No

3. Has the statistical analysis been performed appropriately and rigorously?

Reviewer #3: No

4. Have the authors made all data underlying the findings in their manuscript fully available (please refer to the Data Availability Statement at the start of the manuscript PDF file)?

The PLOS Data policy requires authors to make all data underlying the findings described in their manuscript fully available without restriction, with rare exception. The data should be provided as part of the manuscript or its supporting information, or deposited to a public repository. For example, in addition to summary statistics, the data points behind means, medians and variance measures should be available. If there are restrictions on publicly sharing data—e.g. participant privacy or use of data from a third party—those must be specified.requires authors to make all data underlying the findings described in their manuscript fully available without restriction, with rare exception. The data should be provided as part of the manuscript or its supporting information, or deposited to a public repository. For example, in addition to summary statistics, the data points behind means, medians and variance measures should be available. If there are restrictions on publicly sharing data—e.g. participant privacy or use of data from a third party—those must be specified.

Reviewer #3: No

5. Is the manuscript presented in an intelligible fashion and written in standard English?

Reviewer #3: No

6. Review Comments to the Author

Reviewer #3: Review comments: Magnitude of utilisation of recommended healthcare waste disposal practices at healthcare service delivery points in Uganda

Title: Consider revising to ‘Assessing the levels of utilisation of xxxx recommended healthcare waste disposal practices at healthcare service delivery points in Uganda

mention the specific recommendation xxxx, e.g., WHO-recommended, etc.

Author affiliations: Write this in the same pattern as the correspondent author’s information, with the P. O. Box address, Names of Department, School or Faculty or College, University, Town, and City.

Abstract:

Lines 39-40: “Inadequate healthcare waste management (HCWM) poses significant challenges to public health, environmental sustainability and community wellbeing”. Comment: considers replacing the word ‘significant’ with ‘substantial’ since the former is heavily linked with p-values less than 0.05 and is more statistical than practical. Consider this suggestion throughout the manuscript.

Lines 43-44: “We assessed the level of use of recommended healthcare waste disposal practices at healthcare service delivery points (HSDPs) in Uganda”. Comment: Anytime you use ‘recommended healthcare waste disposal practices’, readers would like to know which specific recommendation you are referring to. It is important to state this upfront and throughout the manuscript. The use of HSDPs is not a standard abbreviation; writers may rather consider using ‘healthcare waste disposal practices’ (HWDP) at ‘healthcare service delivery points’ (HSDP) in Uganda.

Lines 46-47: “We conducted a cross-sectional survey among 681 HSDPs randomly selected from the 15 sub-regions of Uganda between September and December 2023”. Comment: Remove the article ‘the’ since readers may not already be aware of those ‘15 sub-regions…’

Lines 49-50: “The sample size of the HSDPs was determined using the standard sampling methodology provided by the United Nations Population Fund (UNFPA)”. Comment: Readers would like to know the actual sample size calculated, and this is stated within the abstract.

Lines 55-57: “We found notable variations in the use of recommended HCW disposal practices, for instance, the use of incineration and centralised collection by a special agency across different facility levels and regions”. Comment: What does ‘notable’ mean? Is there a value from the findings depicting ‘notable’? This must be stated.

Lines 65-66: “Additionally, urban facilities were more likely to use regular garbage disposal (59.4%, 149/251) compared to rural facilities (19.5%, 83/425)”. Comment: It will be critical to state the magnitude of likelihood of regular garbage disposal practices in urban facilities in comparison with rural counterparts, along with the corresponding p-values. This will mean conducting further inferential analysis, like regression, to determine odds ratios, etc.

Lines 68-72: “The findings from this nation-wide assessment of healthcare disposal practices highlight critical gaps in the use of recommended healthcare waste disposal practices at HSDPs in Uganda, with large variations by facility level, location, and region. These gaps require targeted interventions, including the review and enforcement of existing guidelines and regulations on healthcare waste management”. Comment: It will be important for the study conclusions to align with the stated objectives and key findings, as this is currently not the case.

Methods

Comments: Information on study participants and their characteristics needs to be provided. Who were the ‘HSDP’ managers?. An additional strength of this study will be to identify covariates and outcome variables and inferentially measure associations between key variables, thereby broadening its appeal and strengthening its rigour. This is because using only descriptive analysis narrows the scope of this journal. Providing the study site and locations in a global perspective by writing the GPS coordinates and including maps that link the selected study settings will be more appropriate.

Results

The background and demographic features of the 681 of 762 ‘HSDPs’ surveyed need to be presented as Table 1, segregated by sex, age, location, etc., and the details used to ascertain possible effects on waste disposal practices, etc.

Discussion and Conclusions

Will then be revised to reflect the current findings.

7. PLOS authors have the option to publish the peer review history of their article (what does this mean?). If published, this will include your full peer review and any attached files.). If published, this will include your full peer review and any attached files.

**Do you want your identity to be public for this peer review?** For information about this choice, including consent withdrawal, please see our Privacy Policy..

Reviewer #3: No

Figure Resubmissions:

---

## [Decision Letter · Decision Letter 3]

24 Mar 2026

Assessing the levels  of utilisation of WHO-recommended healthcare waste disposal practices at healthcare service delivery points in Uganda

PGPH-D-25-00169R3

Dear Dr. Kayemba,

We are pleased to inform you that your manuscript 'Assessing the levels  of utilisation of WHO-recommended healthcare waste disposal practices at healthcare service delivery points in Uganda' has been provisionally accepted for publication in PLOS Global Public Health.

Best regards,

Julia Robinson

Executive Editor

Reviewer Comments (if any, and for reference):

Reviewer's Responses to Questions

**Comments to the Author**

1. If the authors have adequately addressed your comments raised in a previous round of review and you feel that this manuscript is now acceptable for publication, you may indicate that here to bypass the “Comments to the Author” section, enter your conflict of interest statement in the “Confidential to Editor” section, and submit your "Accept" recommendation.

Reviewer #3: (No Response)

2. Does this manuscript meet PLOS Global Public Health’s publication criteria? Is the manuscript technically sound, and do the data support the conclusions? The manuscript must describe methodologically and ethically rigorous research with conclusions that are appropriately drawn based on the data presented.? Is the manuscript technically sound, and do the data support the conclusions? The manuscript must describe methodologically and ethically rigorous research with conclusions that are appropriately drawn based on the data presented.

Reviewer #3: (No Response)

3. Has the statistical analysis been performed appropriately and rigorously?

Reviewer #3: (No Response)

4. Have the authors made all data underlying the findings in their manuscript fully available (please refer to the Data Availability Statement at the start of the manuscript PDF file)?

The PLOS Data policy requires authors to make all data underlying the findings described in their manuscript fully available without restriction, with rare exception. The data should be provided as part of the manuscript or its supporting information, or deposited to a public repository. For example, in addition to summary statistics, the data points behind means, medians and variance measures should be available. If there are restrictions on publicly sharing data—e.g. participant privacy or use of data from a third party—those must be specified.requires authors to make all data underlying the findings described in their manuscript fully available without restriction, with rare exception. The data should be provided as part of the manuscript or its supporting information, or deposited to a public repository. For example, in addition to summary statistics, the data points behind means, medians and variance measures should be available. If there are restrictions on publicly sharing data—e.g. participant privacy or use of data from a third party—those must be specified.

Reviewer #3: (No Response)

5. Is the manuscript presented in an intelligible fashion and written in standard English?

Reviewer #3: (No Response)

6. Review Comments to the Author

Reviewer #3: It seems the authors have addressed the concerns raised in the initial manuscript; however, the resubmitted revision appears distorted in the portal, making this verification difficult.

7. PLOS authors have the option to publish the peer review history of their article (what does this mean?). If published, this will include your full peer review and any attached files.). If published, this will include your full peer review and any attached files.

**Do you want your identity to be public for this peer review?** For information about this choice, including consent withdrawal, please see our Privacy Policy..

Reviewer #3: No
